



# Sub-seasonal thaw slump mass wasting is not consistently energy limited at the landscape scale

Simon Zwieback[1,2], Steven V. Kokelj[3], Frank Günther[4], Julia Boike[4], Guido Grosse[4], and
Irena Hajnsek[2,5]

[1]Department of Geography, University of Guelph, Guelph, Canada
[2]Department of Environmental Engineering, ETH Zurich, Zurich, Switzerland
[3]Northwest Territories Geological Survey, Government of Northwest Territories, Yellowknife, Canada
[4]Periglacial Research, Alfred Wegener Institute, Potsdam, Germany
[5]Microwaves and Radar Institute, German Aerospace Center (DLR), Wessling, Germany

*Correspondence to:* Simon Zwieback (zwieback@uoguelph.ca), Irena Hajnsek (hajnsek@ifu.baug.ethz.ch)

**Abstract.** Predicting future thaw slump activity requires a sound understanding of the atmospheric drivers and geomorphic controls on mass wasting across a range of time scales. On sub-seasonal time scales, sparse measurements indicate that mass wasting at active slumps is often limited by the energy available for melting ground ice, but other factors such as rainfall or the formation of an insulating veneer may also be relevant. To study the sub-seasonal drivers, we derive topographic changes from single-pass radar interferometric data acquired by the TanDEM-X satellite (12 m resolution). The high vertical precision (around 30 cm), frequent observations (11 days) and large coverage (5000 km2) allow us to track volume losses as drivers such as the available energy change during summer in two study regions. We find that thaw slumps in the Tuktoyaktuk coastlands, Canada, are not energy limited in June, as they undergo limited mass wasting (height loss of around 0 cm/day) despite the ample available energy, indicating the widespread presence of an insulating snow or debris veneer. Later in summer, height losses generally increase (around 3 cm/day), but they do so in distinct ways. For many slumps, mass wasting tracks the available energy, a temporal pattern that is also observed at coastal yedoma cliffs on the Bykovsky Peninsula, Russia. However, the other two common temporal trajectories are asynchronous with the available energy, as they track strong precipitation events or show a sudden speed-up in late August, respectively. The observed temporal patterns are poorly related to slump characteristics like the slump area. The contrasting temporal behaviour of nearby thaw slumps highlights the importance of complex local and temporally varying controls on mass wasting.

## 1 Introduction

Thaw of ice-rich permafrost, termed thermokarst, has widespread impact on terrain, local ecosystems and the global climate, but the processes that control its abundance and rates remain poorly understood (Grosse et al., 2011; Kokelj et al., 2015). High-frequency observations of terrain modification are necessary to elucidate the drivers of thermokarst and to develop physically-based models to simulating the impacts of permafrost degradation in a changing climate (Lewkowicz, 1987; Günther et al., 2015). The sub-seasonal dynamics of thermokarst are also important because they have a direct impact on the local hydrology,





**Thaw slumps in the two study areas**

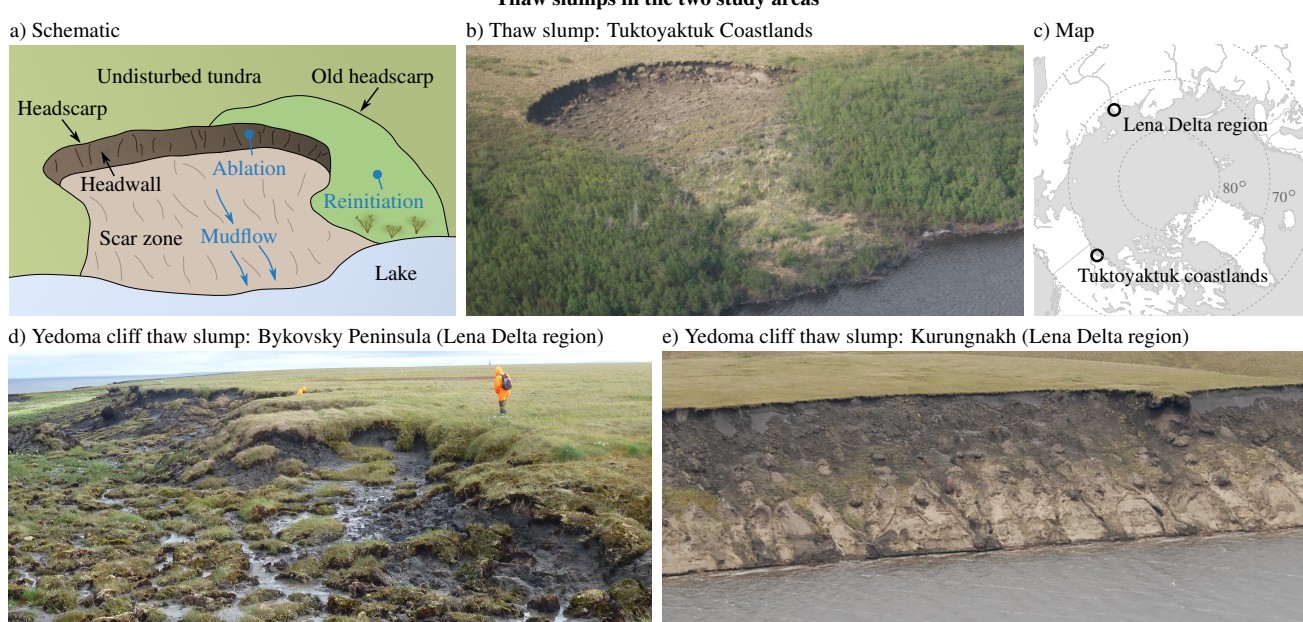

**Figure 1.** a) Schematic of a thaw slump labelling features in black and processes in blue. b) Retrogressive thaw slump in the Tuktoyaktuk coastlands surrounded by relict, now densely vegetated scars on either side (by S. Zwieback). c) Map of the study regions. d) Headscarp of an elongated cliff thaw slump along Bykovsky's coast (by G. Grosse). e) Thaw slump along the east coast of Kurungnakh (by J. Boike). Note the clearly visible ice-poor sand unit in the lower half of the approximately 30 m-high exposure down to the river level.

biogeochemistry and sediment budget (Bowden et al., 2008). In particular, the seasonal timing and magnitude of the thaw-induced mobilization of organic carbon and nutrients influence their lateral transport and chemical fate, and hence the type and magnitude of greenhouse gases released (Grosse et al., 2011; Vonk and Gustafsson, 2013).

Rapid permafrost degradation in ice-rich regions is associated with characteristic landforms. Depending on the topographic
5    position, these are shaped by a wide range of physical processes which we here include under the umbrella term thermokarst. In flat to gently rolling terrain, thermokarst can be closely coupled to changes in local hydrological conditions, with impoundment of water leading to the formation of thermokarst lakes and wetlands, thermokarst pits and many more landforms (Jorgenson, 2013). On hillslopes and steeper terrain, water also plays a key role in initiating or enhancing thaw degradation in landforms such as thermo-erosional gullies, coastal and riverbank bluffs, and thaw slumps (Lacelle et al., 2015; Kokelj et al., 2009).
10    Here, we focus on thaw slumps in which icy sediments are exposed at a steep ice-rich headwall (Fig. 1). They also comprise a low-angled scar zone or slump floor. Thaw slumps commonly develop adjacent to streams, lakeshores or coastlines where thermal, fluvial and coastal erosion can initiate these disturbances by exposing ice-rich permafrost (Jorgenson, 2013; Lantuit and Pollard, 2008).

Thaw slumps are shaped by thermal, hydrological and mechanical processes over their entire life cycle from initiation to
15    stabilization. Once initiated, active thaw slumps can enlarge by several metres per year (Lantuit and Pollard, 2008) as the



headwall retreats into the upslope terrain (Fig. 1). Headwall retreat is linked to processes in the scar zone. The mass wasting at the headwall releases sediment, which has to be removed via the scar zone in order for backwasting to proceed unabatedly (Kokelj et al., 2015). If accumulated material insulates the ice-rich permafrost or if the headscarp retreats into ice-poor terrain, the thaw slump can stabilize (Lacelle et al., 2015). The transport and accumulation of sediment is coupled to the hydrological

conditions, as meltwater and thawing debris typically form a saturated slurry in the slump scar zone (Lantuit and Pollard, 2008). Depending on the water content as well as the sediment input, slope and material properties, this can be a zone of net accumulation or of net volume loss. Net accumulation occurs when the sediments cannot be removed sufficiently quickly: close to the headwall, buttresses of accumulated material can protect the ground ice and reduce retreat rates (Lacelle et al., 2015). Conversely, downslope removal of thawed material at the foot of the headwall can accelerate retreat by exposing ice-rich soil

and by increasing the local relief (Kokelj et al., 2015). Thermal processes in the scar zone can also help sustain thaw slump activity by effecting height losses, which are caused by the melting of ground ice in the warm scar zone (Burn, 2000).

The most important processes in driving headwall mass wasting is the ablation of ground ice. To melt the ground ice at the aerially exposed headwall, energy is required. Ablation increases with insolation and air temperature, as the key terms in the surface energy balance are the radiation input and the turbulent exchange with the atmosphere (Lewkowicz, 1987). If ablation

is the rate-limiting process, the sub-seasonal rates of volume losses will track the incoming radiation and air temperature. On sub-seasonal timescales of days to weeks, the temporal signature is typically steady and slowly declining towards the end of summer (Lewkowicz, 1987). Despite the recognized importance of this process, the prevalence of energy-limited mass wasting has not been assessed at the landscape scale, thus limiting the skill of current thermokarst predictions.

Headwall mass wasting is not always energy limited, and such conditions may be detected using observations of sub-seasonal

volume losses. One such exception occurs when an insulating veneer protects the ground ice, thus slowing down volume losses (Kokelj et al., 2015). Such an insulating cover, principally derived from the thawing sediments themselves, as well as from late-lying snow, commonly persists in early summer (Kokelj et al., 2015; Lacelle et al., 2015). Early summer mass wasting may also be subdued because the incoming energy is used to warm the cold permafrost to the melting point before ablation can set in. A separate agent that can govern mass wasting rates is the availability of liquid water from melting ground ice,

snowmelt or precipitation (Lantuit and Pollard, 2008). Intense precipitation events may accelerate mass losses in the headwall area in some slumps, both via the removal of debris on or at the base of the headwall, and by water supplying additional energy (Burn and Friele, 1989; Barnhart, 2013; Kokelj et al., 2015). As the additional water input can also liquify the sediments in the scar zone and induce downslope flow, thus lowering the base level for erosion and facilitating the evacuation of the headwall area, precipitation can also feed back positively on headwall mass losses via scar zone processes. Finally, failure related to

mechanical instabilities is an important mass-wasting process (Lewkowicz, 1987). Mechanical failure is common when the base of a permafrost exposure is temporarily in contact with open water, the strong thermal and mechanical influence of which (thermo-abrasion) leads to undercutting, niche formation and subsequent block failure (Wobus et al., 2011). On coastal cliffs, niche formation is closely tied to open water conditions and sea temperatures. It hence sets in later than energy-limited ablation but can remain effective for longer in autumn (Günther et al., 2015): its prevalence can thus be inferred based on a late-summer

continuation or speed-up in elevation losses. All three processes have been previously observed in field studies, but little is



known about their prevalence and spatial association, as landscape-scale observations of the sub-seasonal mass loss dynamics have up to now not been available.

To study the sub-seasonal dynamics of thaw slump mass wasting, we use synoptic measurements of topographic changes with a nominal temporal repeat period of 11 days. More generally, vertical changes are an excellent proxy for the degrada-

tion of ice-rich permafrost soils because thawing causes the soils to lose cohesion and reduce in volume, inducing slumping and subsidence (Günther et al., 2015; Jones et al., 2013). Our estimates of topographic changes are obtained from repeated topographic observations using the radar remote sensing technique single-pass interferometry (Bamler and Hartl, 1998). By repeated application of single-pass measurements, time series of the topography and hence topographic changes can be derived (Poland, 2014). Repeated observations of two permafrost regions with high ground ice content were made by the TanDEM-X

satellite pair in the Science Phase (June - August 2015). The frequent acquisitions every 11 days, the high precision of 20-40 cm and the planimetric resolution of 12 m make this data set an excellent opportunity to study the sub-seasonal dynamics of rapid permafrost degradation.

Single-pass interferometry is a promising technique for observing thaw-induced topographic changes on the landscape scale. While it has not been employed in permafrost environments, the technology is mature, as evidenced by the widespread use

of the digital elevation models obtained from the shuttle radar topography mission or TanDEM-X, or the application of such data for quantifying temporal changes in volcanology and glaciology (Krieger et al., 2007; Poland, 2014). The ability to cover large areas and to do so frequently are key advantages over both in-situ measurements and the remote sensing techniques photogrammetry and LiDAR (Günther et al., 2015; Jones et al., 2013; Obu et al., 2016). A further advantage is that reliable height measurements can also be made when the soil moisture changes and when the surface structure is disrupted – a common

occurrence in rapid mass movements. This is in contrast to the closely related technique differential radar interferometry, which is capable of providing synoptic estimates of more subtle elevation changes associated with seasonal and secular thaw subsidence (Liu et al., 2015; Zwieback et al., 2016). Single-pass data, by contrast, are typically not sensitive enough to capture these slow processes over yearly time scales, but instead are ideal for more rapid thermokarst phenomena.

Here, we pursue two objectives:

– to derive synoptic estimates of topographic changes and their uncertainty in two ice-rich permafrost regions in the Northwest Territories, Canada, and in the Sakha Republic, Russia (around 5000 km$^2$) using TanDEM-X data acquired during the Science Phase in 2015

– to analyse the sub-seasonal dynamics of topographic changes at slump headwalls and their variability between features with the aim of attributing the observed patterns to potential drivers

Our guiding hypothesis is that the volume losses are governed by the ablation of ice, and hence limited by the available energy. To test this hypothesis on time scales of one to several weeks, we compare the observed temporal dynamics to the energy available for melting ice, which we estimate using a model driven by ground measurements and additional satellite data. Deviations from energy-limited dynamics are common during the entire thaw period. To attribute these deviations to additional



controls, we compare the sub-seasonal fluctuations of mass losses to potential drivers such as precipitation and insulation by snow based on the distinct temporal signatures of these drivers.

## 2 Study areas

The two study areas are underlain by continuous ice-rich permafrost, and they are locally affected by hillslope thermokarst. The Tuktoyaktuk coastlands in the Mackenzie Delta region in the Northwest Territories, Canada (Fig. 1), are a glacially shaped landscape that contains areas rich in massive ground ice, where retrogressive thaw slumping is common along lake shores (Burn and Kokelj, 2009). Conversely, the Lena River Delta area, northern Yakutia, Russia, was not glaciated. Our data cover two sites in this region, both of which are characterized by extensive yedoma uplands that are underlain by fine-grained, ice-rich Pleistocene deposits. Their very high total ground ice content of up to 90% by volume makes them vulnerable to rapid coastal and river bank erosion and thaw slumping (Wetterich et al., 2008). In summary, the two study areas provide contrasting climatic conditions, geological histories, and geomorphological processes for analysing the sub-seasonal dynamics of thaw slump activity.

The Tuktoyaktuk coastlands are covered by two crossing TanDEM-X orbits that mainly extend in the north-south direction (>100 km), yielding a total area of 4500 km$^2$. The north-south extent transitions from the boreal forest at Inuvik in the south to the dwarf shrub tundra at Tuktoyaktuk on the Beaufort Sea coast. This ecological transition is associated with pronounced gradients in climate (Burn and Kokelj, 2009). The southern part of the study area is about 3 degrees warmer in summer than the north (mean July temperature of 14.1 °C in Inuvik vs 11.0 °C in Tuktoyaktuk (can, 2017), whereas the temperature is more uniform in winter. Annual precipitation decreases from 240 mm in the south to 161 mm in the north. The vegetation reflects the climatic gradient, as forest transitions to shrub tundra. The transition zone is characterized by a northward decrease in the density and height of tall shrubs. The gradients in climate and vegetation combine to shape the ground thermal regime, as the minimum near-surface ground temperature decreases from about -3°C in the south to -7°C in the north (Kokelj et al., 2017b).

Retrogressive thaw slumps can be abundant where the relief position and surficial geology are favourable (Fig. 1). They almost exclusively occur in immediate proximity to tundra lakes, which are widespread in most geological units within the study area (Kokelj et al., 2009). The surficial geology varies from hummocky moraine in the south to an increasing proportion of lacustrine plains in the north, interspersed with ice-rich hummocky moraine and glaciofluvial deposits, both of which may host massive ground ice (Aylsworth et al., 2000). The hummocky morainal materials deposited at the margins of the Laurentide ice sheet are highly susceptible to thaw slumping (Kokelj et al., 2017a). Thaw slumps can grow to areas exceeding several hectares. Headwall heights can reach up to about 15 m, depending on geology and topography (Kokelj et al., 2009). In addition to the mass wasting at thaw slumps, areas of notable slope erosion also occur along the Beaufort Sea coast (Wolfe et al., 1998), and in ice-poor sandy bluffs exposed at large water bodies such as the Eskimo lakes.

The second study area, the Lena River Delta area in north-east Siberia, consists of two sites (Fig. 1). The first site, the Bykovsky Peninsula, is located southeast of the delta close to the harbour town of Tiksi on the Laptev Sea coast. The climate is subpolar, with mean monthly temperatures varying from -31 °C in February to 8 °C in July in Tiksi (Günther et al., 2015).



Geologically, it is characterized by very ice-rich yedoma uplands consisting of thick Pleistocene deposits, interspersed with thermokarst lakes and drained thaw lake basins (alases) (Grosse et al., 2007; Schirrmeister et al., 2017). The Bykovsky Peninsula is subject to continual coastal erosion both along yedoma and the alas coasts (Lantuit et al., 2011). Yedoma uplands that are exposed along the coast form elongated retrogressive thaw slumps (Günther et al., 2013). The upper part of these bluffs, whose

height can exceed 40 m, consists of a vertical icy headwall (Fig. 1d). Below the slump headwall, the slopes are more graded but still comparatively steep, shaped by the balance between sediment supply and removal along the bluff. Apart from along the coast, rapid permafrost degradation occurs along lake shores, within gullies, or where vehicle traffic and other disturbances induce localized thermokarst.

Kurungnakh Island is located in the southern central Lena River Delta, Russia, around 120 km west of the Bykovsky Penin-

sula. Its location further inland is associated with slightly colder February air temperatures (-33 °C) and slightly warmer July air temperatures (10 °C, Boike et al. (2013)). The part of Kurungnakh we focus on is also largely covered by yedoma sediments (Morgenstern et al., 2011). Along its eastern margin, bordering the Olenyekskaya Channel, the yedoma sediments (around 25-30 m thick) and the underlying ice-poor fluvial sands form steep cliffs of up to 40 m height above river level (Wetterich et al., 2008; Kanevskiy et al., 2016), see Fig. 1e. In addition, thaw slumping also occurs on slopes surrounding to thermokarst lakes

within alases (Morgenstern et al., 2011).

## 3  Methods & Data

### 3.1  Height changes and rates from interferometry

#### 3.1.1  Estimating height changes

TanDEM-X bistatic image pairs acquired during the Science Phase 2015 (June to August) served as input for the topographic

mapping. The image pairs were acquired with particularly large across-track baselines corresponding to heights of ambiguity of 8–14 m, with which height precisions of better than 0.5 metres can be achieved (Tab. S1). The topographic information was derived from the Coregistered Single-look Slant-range Complex (CoSSc) products. They have a native planimetric resolution of around 3 m, depending on the study area (Tab. S1), which was reduced to 10-12 m during the interferometric processing.

Estimates of topographic changes $\Delta h$ were derived from time series of TanDEM-X CoSSc image pairs. Rather than fol-

lowing the standard approach of first deriving a digital elevation model from each image pair interferogram and subsequently forming the differences (Poland, 2014), we directly differenced interferograms. Our rationale was that the baselines were essentially constant for all acquisitions, so that the differencing of interferograms yielded a direct estimate of $\Delta h$ and phase unwrapping was greatly facilitated.

The interferogram for every image pair was formed from the CoSSc product by standard methods (range spectral filtering,

removal of the flat earth/topographic phase from the input DEM, multilooking) and this time series was co-registered (Bamler and Hartl, 1998). Subsequent co-registered interferograms $m$ and $n$ were differenced by forming the phase difference $\Delta\phi =$





$\phi_n - \phi_m$ which contains the required information about the height difference $\Delta h = h_n - h_m$

$$\Delta \phi = k_{z,n} h_n - k_{z,m} h_m + \phi_{\mathrm{mov},mn} + \phi_{\mathrm{offset},mn}$$
$$= k_{z,n} \Delta h - (k_{z,m} - k_{z,n}) h_n + \phi_{\mathrm{mov},mn} + \phi_{\mathrm{offset},mn} \qquad (1)$$

$\Delta h$ is related to this observable via the vertical interferometric wavenumber or the height sensitivity $k_{z,n}$ of interferogram

$n$. The interferometric wavenumber can be derived from the orbit data and an auxiliary input DEM with sufficient accuracy (Rizzoli et al., 2012). In order to estimate $\Delta h$, the other terms had to be quantified and removed. The second term is a small residual topographic contribution that we removed using the auxiliary DEM. The third term $\phi_{\mathrm{mov},mn}$ due to the along-track baseline is zero over dry land but can be non-zero over moving water surfaces (Fig. S2). The fourth term $\phi_{\mathrm{offset},mn}$ is a phase offset e.g. due to orbital errors which changes only slowly across the interferogram. We removed it by mild high-pass

filtering (Rizzoli et al., 2012; Poland, 2014) with a length-scale of 600 m that is much larger than the individual thermokarst disturbances. The filtering procedure was robust to outliers as it was based on the median and was not applied to masked pixels like radar shadow or water surfaces. Note that this filtering also largely cancelled isotropic seasonal subsidence, which we will generally neglect in the following because of its small magnitude compared to the uncertainties.

### 3.1.2    Uncertainties

The uncertainty in the observed height changes was derived from the phase noise, which in turn was estimated from the observed coherence magnitude $|\gamma|$. The coherence magnitude is an indicator of the similarity of the image pair: it takes on values between 0 (no similarity) and 1 (perfect similarity and high phase quality) (Bamler and Hartl, 1998). We employed standard techniques to translate the phase noise inferred from the observed coherence to an uncertainty in $\Delta h$ using the Cramer-Rao approximation (see Sec. S1.1). These uncertainties were typically between 30-60 cm, but they varied as they

depended on the imaging geometry (baseline, incidence angle) and on the spatially variable coherence magnitude.

The coherence magnitude and with it the phase noise are influenced by surface characteristics and measurement properties in several ways. Firstly, a loss of coherence can be induced by the additive measurement noise (Krieger et al., 2007). Secondly, temporal changes between the two acquisitions also reduce the coherence. While this effect is minimized in the single-pass system TanDEM-X, a short effective temporal baseline remains, which is associated with decorrelation over water surfaces.

The decorrelation is expected to increase with wind speed: simple modelling further indicated that it may also be relevant over mixed pixels that contain sub-resolution water bodies (Fig. S2a). Finally, the height variability within the resolution cell is associated with geometric decorrelation (Fig. S2b). For extended planar slopes, this can be largely compensated for by spectral filtering, but the impact of vegetation and irregular terrain cannot be cancelled (Krieger et al., 2007). Vegetation also biases the height estimates, i.e. the estimated height will not coincide with the terrain height; we will assess the impact on estimating $\Delta h$

in the shrub tundra separately when we consider measurement biases.

The coherence-based uncertainty estimates were assessed independently and found to be accurate to within approximately 30% and generally conservative (see Sec. S1.3 for details). The rationale of the assessment was to compare the predicted uncertainty to the observed variability within areas that could be assumed stable and homogeneous (Rizzoli et al., 2012). The



analysis of stable areas further allowed us to assess residual biases due to $\phi_{\text{offset}}$, which were found to be small compared to the overall uncertainty (around 2 cm). Also the uncertainty due to errors in the input DEM or the orbit information was estimated to be small by comparison. In other words, the phase noise is the limiting factor in the precision of estimated height changes.

The observed height change does not necessarily reflect the true height change within the resolution cell. We found three important sources of bias in the tundra: late-lying snow packs, shrub phenology and water surfaces. The ablation of snow packs induced an apparent lowering of the surface (negative observed elevation changes) of more than 1 m (Fig. S1), which could be mistaken for thermokarst. Conversely, over tall shrubs the single-pass observations indicated positive elevation changes of several decimetres in June, coincident with leaf-out (C. Wallace, personal communication). Finally, over water bodies, the sign of the measurement bias depended on the wind conditions, as predicted in Fig. S2. This error source was the largest, as its magnitude exceeded several metres. We believe that the best way to mitigate all these biases is to mask them where necessary. For instance, our focus here is on hillslope thermokarst phenomena, so that open water surfaces were of minor concern. By contrast, snow was a major error source, at least in June, when late-lying snow patches persisted in many slumps. To assess the role of snow on the measurements, we mapped the presence of snow patches within slumps using medium-resolution satellite imagery in June and July. The details of this analysis, as well as an in-depth assessment of biases can be found in Sec. S1.2.

### 3.1.3 Estimating average rates and their uncertainties

Time-average elevation loss rates were computed by stacking time series of individual $\Delta h$ measurements. The stacked elevation loss rate $r_s$ was estimated in a generalized least-squares procedure that accounted for the uncertainty of the individual observations (see Sec. S1.4 for mathematical details). It will typically be reported in cm/day, as will be elevation loss rates $r$ between two acquisitions.

### 3.2 Mapping of disturbances

To identify and map disturbances in the study regions, we used high to medium resolution optical data. In the Tuktoyaktuk coastlands, we inventoried 160 thaw slumps that showed signs of recent activity with Sentinel-2 imagery from 2016 (10 m visible and near infrared). The slumps were identified based on their distinctive appearance caused by exposed mineral soils and limited vegetation cover, their shape and the presence of a headscarp (Lantuit and Pollard, 2008). In polycyclic slumps, i.e. when slumping activity had re-initiated within an older slump, we mapped those units that showed signs of recent activity. Within the inventory, all slumps were located in immediate proximity to lakes and their distribution was non-uniform, with higher abundance on morainal deposits (Kokelj et al., 2009). The slump size varied by about two orders of magnitude and could reach up to more than 5 ha (Fig. S8); 14% of the slumps exceeded 1 ha in area, but the median size was considerably smaller (0.4 ha). Slopes of all aspects are affected by slumping, but features with northeast and northwest orientations are more common, which is consistent with previous findings by (Kokelj et al., 2009). To further characterize the slumps, we extracted diverse attributes from the satellite image (Normalized Difference Vegetation Index: NDVI) and from topographic data sets (elevation and drainage for the slump centroid from a pre-disturbance DEM, local relief as a proxy for headwall height from the TanDEM-X data). To quantify the decadal-scale dynamics, we analysed orthophotos from 2004 (<1 m resolution). For each





slump (except one, which was not covered by the aerial imagery), we determined whether the location had not been disturbed in 2004 (14% of the slumps), whether the same slump had already been present and had continued activity (21%), or whether there had been an earlier generation slump (65%). During this time interval, the areal expansion of the slumps was 0.3 ha on average for the latter two categories. In addition to the thaw slumps, we also mapped several disturbed shorelines along the

Eskimo Lakes, the largest lake in the study region. On the Bykovsky Peninsula and on Kurungnakh Island, we mapped active elongated coastal/riverbank thaw slumps as well as thaw slumps along lake shores.

To quantify the thaw-induced volume losses, we manually delineated active areas within the previously identified thaw slumps in the TanDEM-X data. Despite the clear signal of change in the headwall area of active slumps (Fig. S9), constraints associated with the 12 m resolution of the TanDEM-X data have to be considered. The mapped resolution cells may thus also

contain undisturbed terrain and scar zone surfaces on either side of the ablating headwall. Thaw slumps were labelled inactive when no subsidence could be detected along the headwall. For the active landforms, representative height changes and stacked elevation loss rates $r_s$ were computed by aggregating the TanDEM-X resolution cells within these active parts and forming the median. The uncertainty of the aggregate elevation loss rate $r_s$ was estimated from the pixel-level uncertainties using parameteric bootstrap analysis (Davison and Hinkley, 1997). Its objective is to mimic the measurement process by generating

many potential data sets (aggregate rate estimates) from which the standard error can be estimated. The measurement process of the aggregate rate depends on the pixel-level TanDEM-X height changes, whose uncertainties in turn we parameterized by their coherence (see Sec. S1.1). To explore the activity and the subsidence rates of all the inventoried thaw slumps as a function of potential controls such as aspect, we employed logistic regression (activity) and standard regression analysis ($r_s$). In addition to mapping areas of volume loss along slump headwalls, we also delineated active areas within the scar zones where

volume changes could be detected in the TanDEM-X data. These scar zone changes typically indicate an elevation gain (Fig. S9), but they were difficult to map because they were generally an order of magnitude smaller than those along the headwall. To compare thaw slump activity with other volume losses, we also estimated volume losses at retreating ice-poor bluffs along the Eskimo Lakes, Tuktoyaktuk coastlands.

### 3.3 Time-series analysis of sub-seasonal dynamics

To explore and interpret the sub-seasonal dynamics of height changes, we used observations of meteorological parameters. In the Tuktoyaktuk coastlands, we analysed in-situ measurements from Inuvik (precipitation, air temperature, humidity, wind speed and air pressure) and Trail Valley Creek (also incoming and outgoing long and shortwave radiation). Both sites are located in the southern part of the study area at a distance of 45 km. Despite the proximity, the precipitation records did not match well as persistent and large deviations are common ($\approx$ 20 mm per week, or up to 100%; see Fig. S10). Radiation

fluxes were also taken from the Ceres SYN1deg-3Hour Ed3A product (incoming shortwave radiation partitioned into direct and diffuse terms), which contains flux estimates from a radiative transfer model and satellite observations (CERES Science Team). The total incoming shortwave radiation and the net longwave radiation compared well with the in-situ measurements (Fig. S10). The Ceres flux data were also employed in the Bykovsky study area, where they were complemented by in-situ meteorological measurements from Tiksi (WMO 21824, 20 km from the study area).



To test the hypothesis that the observed volume losses were energy limited, we estimated headwall ablation using the semi-empirical approach by Lewkowicz (1987). This approach uses meteorological forcing data to estimate the energy available for melting the ground ice by regression formulae that approximate the energy balance modelled using the gradient method for the turbulent exchange (input in-situ measurements: air temperature, water vapour pressure, wind speed) and the net radiation

(derived from Ceres). The net shortwave radiation was estimated as a function of surface slope and aspect by considering the diffuse and direct incoming radiation and using an albedo of 0.15 for the disturbed slump surface (Lewkowicz, 1987). The hourly estimates of the ablation flux (available energy) were aggregated to estimate the total ablation between two subsequent TanDEM-X acquisitions (typically 11 days), which could then be compared to the observations. The model predictions have previously been found accurate at daily to monthly time scales, but the model overpredicted the ablation when the ice face

was partially covered by debris/snow (Lewkowicz, 1987) in early summer. This early season bias was likely exacerbated by the model's lack of a conductive subsurface heat flux term: the entire ground heat flux is used to melt the ice according to the model, whereas in reality part of the heat flux will warm the cold permafrost. The model further does not consider heat transfer from liquid water, and the representativeness of the forcing data was difficult to ascertain (e.g. unmodelled shading effects, variation of meteorological conditions across the study area). The modelled ablation is sensitive to the ground ice

content, which is generally difficult to obtain and also varies across a single slump; we used the value by Lewkowicz (1987). The impact of this assumption was, however, considered small because we focused on the relative temporal variability of the ablation, not its absolute value. The reason for this is that we measured height changes with 12 m resolution, which were expected to be proportional to the ablation, but whose factor of proportionality remained unknown as it depended on the unknown sub-resolution geometry (e.g. how much of the resolution cell was intersected by the headwall). In addition to this

semi-empirical model, we also considered a second reference model of the sub-seasonal dynamics, namely one of uniform rate.

We assessed the consistency of these models with the observations using statistical tests which accounted for the uncertainty of the observations. The null hypothesis of the test was that the time series of height changes were proportional to those predicted by the model. The tests were based on the parametric bootstrapping approach for determining the uncertainties

(Davison and Hinkley, 1997): a sample of potential measurements under the null hypothesis was generated and the p-value was computed by determining how extreme the actual observation was compared to this sample (see Sec. S1.5 for details). Small p-values $p < 0.05$ indicated statistically significant deviations from sub-seasonal dynamics that were either uniform or proportional to the energy balance.

To explore the synchronicity of the sub-seasonal dynamics of elevation losses across the Tuktoyaktuk coastlands, we used

clustering analysis. The fuzzy c-means clustering approach found representative time series (the clusters) so that the normalized dynamics of the landforms within one cluster were as similar, i.e. the volume loss as synchronous, as possible (Liao, 2005). In addition to the clusters, the analysis produced, for each landform, a degree of membership to all cluster centres; we assigned the landforms to the cluster to which they had the highest membership. The number of clusters was determined using the elbow method (see Sec. S1.6 for details).



## 4 Results

### 4.1 Tuktoyaktuk coastlands

In the Tuktoyaktuk coastlands study area, elevation losses were commonly observed in the headwall area of slumps, in contrast to large swathes of the study area which appeared stable. In addition to thaw slumps, we also detected elevation losses at sandy lake-side bluffs along the Eskimo Lakes. Similarly, there was no indication of pronounced gully formation or incision during summer 2015, confirming that in this region thaw slumping is the predominant form of thermokarst-driven landscape change. More than half of the inventoried slumps (89/160; Fig. 2a) exhibited detectable activity in the TanDEM-X data. As the activity and its detectability may be influenced by slump characteristics and the sensor viewing geometry, we compared the detected activity to the slump's NDVI, area and its orientation to the satellite sensor using logistic regression (Fig. S8). Smaller NDVI values, indicating sparse vegetation cover, were associated with a higher probability of detected activity. The model predicted a slightly higher chance of detecting activity when the satellite look direction was parallel to the strike of the headscarp ($c = 0$) than when the headwall was observed from behind ($c = 1$; potential shadow) or face-on ($c = -1$; potential layover). The areal extent of a slump was not a useful predictor of detectable activity (Fig. S8) and neither was the headwall height.

Observed volume loss rates at the active slumps varied throughout the summer season, in a way that did not reflect the energy available for ablation. The discrepancy was most pronounced in early summer (early June to mid-July), as volume losses were smaller than in the second half of summer, despite the ample available energy (Fig. 3; only ascending orbit). The median volume loss rate increased from $0\,\mathrm{cm\,d^{-1}}$ in early June to around $3\,\mathrm{cm\,d^{-1}}$ in late July and August. The acceleration was also evident when looking at the slumps individually as $94\%$ exhibited smaller elevation losses in early June compared to August, indicating the presence of a negative control (e.g. debris cover) on ablation. After all, potential ablation fluxes as predicted using the energy balance approach were highest in June and early July and subsequently dropped by around one quarter. The difference between the sub-seasonal dynamics of the observed volume losses and the hypothesized ablation-driven ones was statistically significant for the majority of slumps ($53\%$; from June to August).

Also in the second half of summer (mid-July to late August), deviations from energy-limited ablation-driven volume losses were commonly observed, as only a subset of the thaw slumps exhibited volume losses that approximately tracked the energy available for ablation. This subset formed one of three distinct categories (as suggested by clustering analysis, see Sec. S1.6), each of which exhibited a large degree of synchronicity. The first cluster C1 corresponded to a fairly steady volume loss, similar to the expected energy-limited trajectory; it contained almost half the slumps (Fig. 4). Conversely, the other two clusters largely exhibited volume losses that did not appear to be controlled by the hypothesized energy-limited variability. Cluster C2 appeared to be related to intense precipitation events recorded at Inuvik in that it showed two peaks in mid-July and mid-August. The degree to which volume losses speeded up during these two time intervals varied across the slumps. As these peaks were at odds with the smoothly varying available energy from turbulent exchange and radiation, the null hypothesis of ablation-driven volume losses could be rejected more often than for features in cluster C1 (Fig. 4; similarly for the hypothesis of uniform dynamics). Finally, less than a quarter of the slumps exhibited an end-of-summer acceleration of subsidence (C3), but





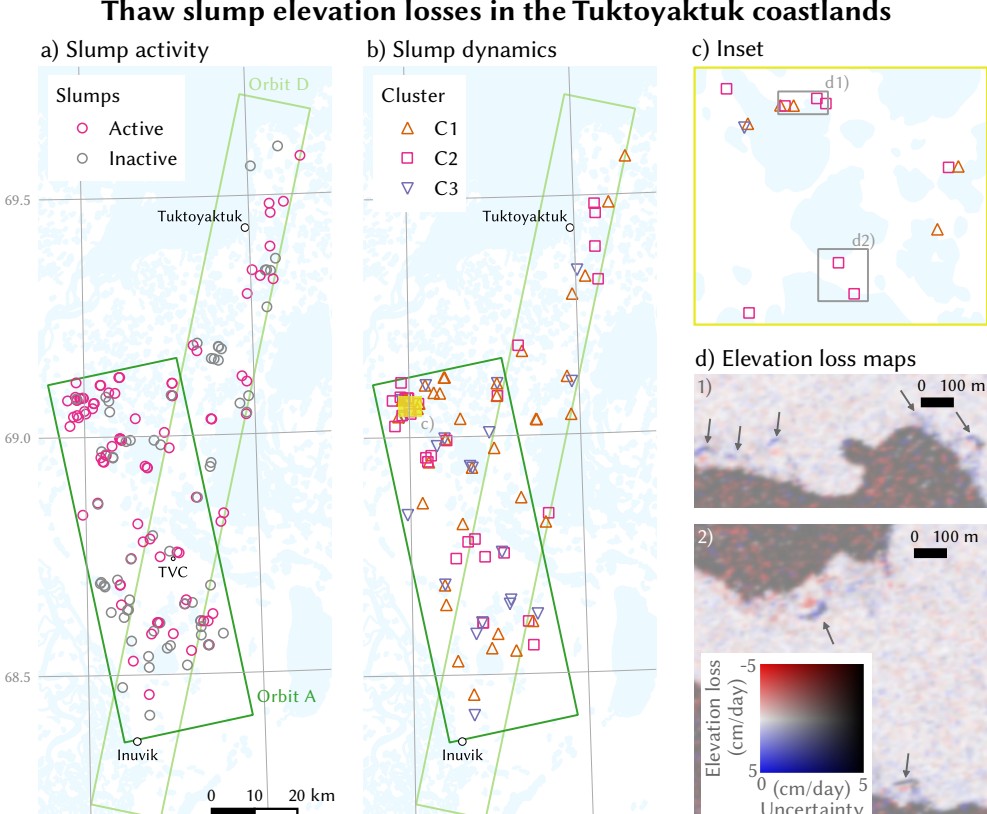

**Figure 2.** Overview of the study area and all mapped thaw slumps. a) Slumps in the TanDEM-X data according to whether activity could be detected. The locations of in-situ measurements at Inuvik and Trail Valley Creek (TVC) are also shown. b) The sub-seasonal dynamics from mid-July to late August form three clusters. c) Inset of the area which is highlighted by a yellow rectangle in b). d) Observed elevation loss rates $r_s$ from mid-July to late August (see grey rectangles in c): elevation loss in blue is evident at arcuate headscarps, elevation gain on some slump floors, high uncertainty water bodies appear in black.





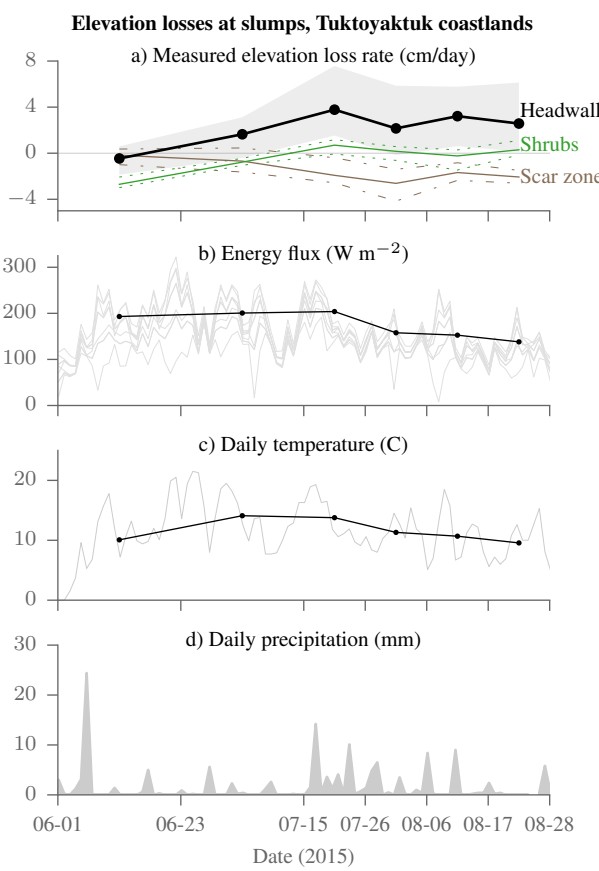

**Figure 3.** Observed time series of elevation losses and meteorological variables in the Tuktoyaktuk coastlands. a) Elevation loss rate at thaw slumps covered by the ascending orbit ($N = 71$; solid black line: median, grey area: interquartile range), along with observed rates over all slump scar zones with detected changes ($N = 25$) and over patches with dense shrub cover ($N = 10$). The observed elevation loss rate is plotted halfway between the earlier and the later acquisition (indicated by ticks). b) Available energy for different headwall orientations (grey) and temporal averages for a horizontal surface (black). c) Temperature at Inuvik (daily averages in grey; averages between subsequent acquisitions in black). d) Daily precipitation measured in Inuvik.





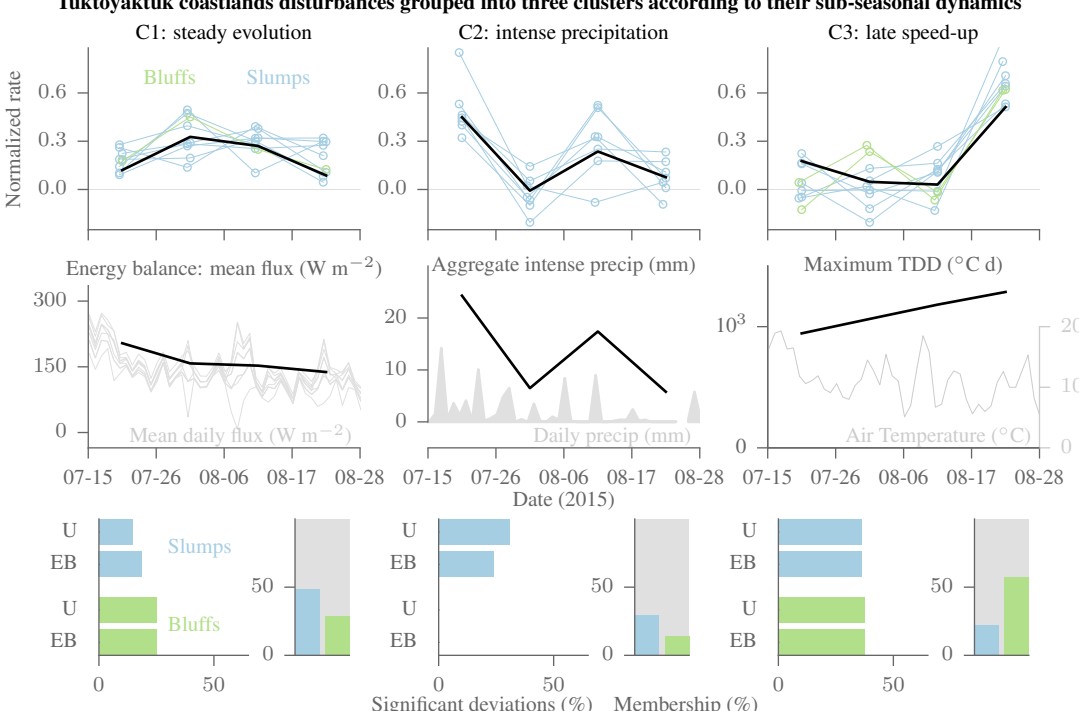

**Figure 4.** Slumps and bluffs in the Tuktoyaktuk coastlands were grouped into three clusters (C1-C3) according to their sub-seasonal elevation loss dynamics after mid-July. The first row contains the normalized rates of three clusters, with the time series of selected slumps and bluffs shown in blue and green, respectively, and the representative cluster dynamics in black. The second row contains the available energy (grey: daily values for different headwall orientation; black: aggregate values for horizontal orientation); the precipitation (daily values measured in Inuvik in grey, aggregate values for days with intense precipitation > 5mm in black); and the air temperature measured in Inuvik (daily values in grey, maximum thawing degree days TDD between subsequent acquisitions in black). The third row contains the percentage of landforms for which the null hypothesis of uniform (U) or energy balance-limited (EB) elevation losses was rejected (horizontal bars) and the percentage of landforms that were classified in the respective cluster (vertical bars).

a majority of ice-poor lake bluffs did (Fig. 4). As the available energy decreased towards late August, the observed accelerating volume losses were predominantly significantly different from the hypothesized energy-limited trajectories.

Not all landforms fitted neatly into this classification, as certain sub-seasonal dynamics appeared to be mixtures of two clusters or different altogether. A few illustrative examples are shown in Fig. S11a, whereas Figs. S12–S17 show all thaw
5 slumps. The examples include slumps that showed intermittent speed-up in late July (classified as C1) or that accelerated slowly during all of August (classified as C3). Also, negative volume losses (apparent uplift) were commonly observed, but their magnitude was rarely larger than the standard error (only for 7%). One example of an unusual slump is the largest of the entire study area, shown in Fig. S11b: it appeared to be a mixture of C1 and C2 in that it speeded up intermittently in mid-August (like C2) but not in early July (unlike C2). This significant acceleration was observed in the measurements from both





orbits and appeared to be limited to around one quarter of the headwall length, illustrating the potential intra-slump variability that the sensor resolution did not allow us to study except for this very large slump.

The cluster membership and hence the sub-seasonal dynamics were poorly related to the geographical location or easily measured slump characteristics such as the size. All three types of dynamics occurred in the entire study area, often in close

proximity (Fig. 2). Neither could they be well distinguished by geometric and topographical properties, as the headscarp aspect, local relief and catchment size were similar for all three clusters (Fig. S18). The slump area and elevation provided some insight, as features in cluster C3 – the late speed-up – tend to be smaller and at slightly higher altitudes. Also a comparison with each slump's state in 2004 – for instance whether the location had been undisturbed – did not reveal any clear-cut relation to the cluster membership. Conversely, detectable summer-time height changes in the scar zone were closely related to the cluster

membership (Figs. S12–S17), as they were chiefly observed at slumps that responded to strong precipitation events (C2). This association, along with the scar-zone elevation change sign (predominant height increase) and magnitude (decimetres), suggests a strong influence of downstream sediment dynamics – as opposed to thaw subsidence – on headwall mass wasting at these slumps.

To further explore the spatial variability of the volume losses in the second half of summer, we also investigated the time-

averaged volume loss rates. Across all slumps this rate $r_s$ varied typically between 2-5 cm d$^{-1}$ with a weak dependence on location and headscarp geometry (Fig. 5a, both orbits). Regression modelling of $r_s$ in terms of slump properties indicated that south-facing slumps were slightly more active (by $\approx 0.8$ cm d$^{-1}$) than north-facing ones (Figs. 5b, S19), which would be consistent with the hypothesized dominance of the available energy (especially insolation). For a given headwall orientation, the elevation loss rates were predicted to increase very little, if at all, with headwall height and slump area. However, the

regression could explain little of the observed scatter ($R^2 = 0.14$) and the regression coefficients tended to be comparable or smaller in magnitude than their standard errors. The inclusion of additional exogenous variables such as accumulation area and latitude improved the fit only marginally and very quickly led to large multi-collinearity. Apart from natural variability and the limited precision of the observations, one likely reason for the lack of spatial consistency was the sub-resolution geometry due to which the observed volume losses were related to the headwall retreat by an unknown and spatially variable factor of

proportionality.

### 4.2 Lena River Delta area

On the Bykovsky peninsula, localized volume losses were observed along the coastal thaw slumps, whereas the interior appeared to be stable. Actively retreating thaw-slump yedoma cliffs were detected on both the east coast (favourable viewing geometry) and the west coast (problematic viewing geometry inducing foreshortening, layover). The mean rates of volume

loss were similar for cliffs on either coast (3 - 5 cm d$^{-1}$, Fig. 6). So were the sub-seasonal dynamics as all volume loss rates were fairly constant from June to August. The near-uniform dynamics resembled the available energy, which changed by only 15% during the summer. However, the similarity of the observations to the hypothesized energy-limited losses may have been misleading in early summer, as the volume losses were likely overestimated before mid-July due to contemporary snow ablation (volume losses were also observed in gullies, and residual snow was still present in early July, see Fig. S20).

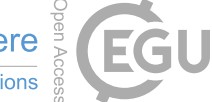



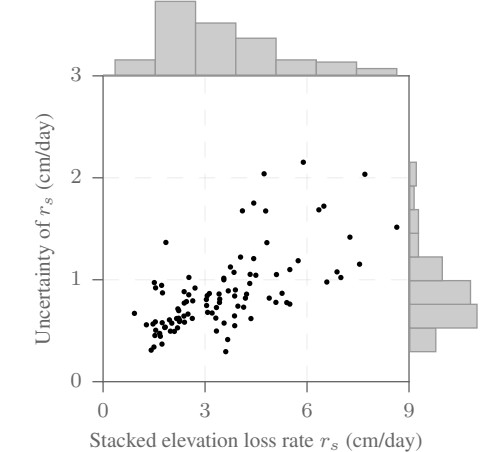

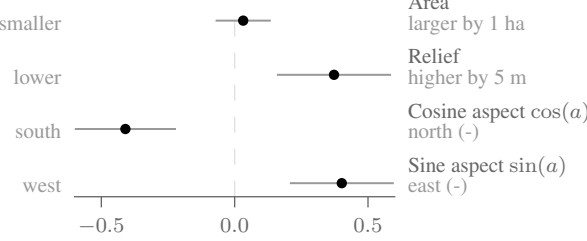

**Figure 5.** Elevation loss rates $r_s$ at slumps (both orbits) in the Tuktoyaktuk coastlands, temporally averaged between 15 July and 28 August 2015. a) Scatter plot of $r_s$ estimates and their standard error. b) Dependence of $r_s$ on slump features as predicted by the regression model: the dots show the predicted change in $r_s$ for a change in the property of the annotated magnitude (e.g. 5 m increase in headwall relief), the bars indicate plus/minus one standard error.

The island of Kurungnakh in the Lena Delta was comparatively stable after the ablation of most snow banks (after 11 July, Fig. S21). The only lake-side retrogressive thaw slump in the area did not show any detectable changes. The steep yedoma river-bank slumps on the eastern shore were very poorly imaged because of extreme foreshortening and layover, the high intensity of which also gave rise to a strong azimuth ambiguity. Consequently, volume losses were detected in only a few spots,

5   despite the known activity of these slumps. The viewing geometry was more favourable at the northern shore, where localized height losses were particularly pronounced in steep gullies. However, we attributed this signal in the gullies largely to snow, as it was most pronounced during mid-July ($\approx 10\text{cm d}^{-1}$) when snow packs were observed to persist in these deep gullies (A. Morgenstern, personal communication). In addition, the water bodies were associated with pronounced spurious height changes (see Fig. S1).



## Topographic changes on the Bykovsky Peninsula

a) False-colour image

b) Elevation loss rates

c) Insets: Yedoma cliffs

d) Time series

**Figure 6.** Rapid permafrost degradation occurred at coastal thaw slumps on the Bykovsky Pensinsula. a) Sentinel-2 image. b) Mean elevation loss from 27 July to 28 August 2015 estimated from TanDEM-X is large at yedoma cliff slumps (rectangles). c) Magnified images showing three coastal stretches from b). d) Time series of observed elevation losses at cliffs and atmospheric conditions similar to Fig. 3, except that the topmost panel shows the dynamics of all cliffs (grey) and their average (black).



## 5   Discussion

Our landscape-scale analyses reveal sub-seasonal patterns of mass wasting that are common to most features, especially the slow onset of ablation in early summer, which suggest the widespread presence of a common control. Conversely, the observed synchronicity of only limited subsets of landforms indicates the presence of distinct processes whose impact is particularly pronounced for only those subsets. The spatial variability only becomes evident in synoptic observations, highlighting the importance of remote sensing approaches for understanding and quantifying permafrost degradation.

The delayed onset of volume losses in early summer despite the large available energy indicates that mass wasting is not energy limited at this time. Lewkowicz (1987) and Lacelle et al. (2015) observed that early-season mass wasting can veneer late lying snowdrifts, protecting ice-rich permafrost from early-season thawing. Snow cover was still widespread in thaw slumps at the time of the first radar acquisition in early June (Tuktoyaktuk coastlands), but likely limited in depth due to the preceding weeks of above-zero temperatures. It had largely disappeared by mid-June (Fig. S22), but subdued ablation persisted into July according to the TanDEM-X data, pointing towards the importance of debris cover, possibly also on top of snow. In addition, even in the absence of a persistent snowbank, a portion of the available energy must also go into warming cold permafrost behind the slump headwall. Conversely, we do not believe the observed early-season signal to be spurious. While snow is also a bias source for radar interferometry (Fig. S1), it cannot spuriously mask actual ablation because the bias is of the wrong sign. By contrast, shrubs that are within a headwall resolution cell may partially mask actual volume losses in early summer due to their phenological development, but it is too small and shrub cover too patchy to explain the reduced volume losses by itself (Sec. S1.2).

Energy-limited mass wasting appears to have been important from mid-July to late August, as a uniform or slowly decreasing activity was typical. Such mass loss driven by the energy available for ablation has previously been found to govern sub-seasonal rates of headwall retreat in Alaska (one slump) and on Banks Island, Canada (three slumps) (Lewkowicz, 1987; Barnhart, 2013). We observed steady mass wasting at the majority of slumps in the Tuktoyaktuk coastlands (C1), and also at coastal slump cliffs on the Bykovsky Peninsula. A strong link between sub-seasonal ablation rates on coastal yedoma cliffs and the energy balance has previously been observed on Muostakh Island (15 km south of Bykovsky Peninsula) (Günther et al., 2015). Our observations of rapid ablation corroborate the notion that when ice-rich permafrost is exposed above ice-poorer units, its loss is the dominant factor in coastal terrain modification (Günther et al., 2013). By contrast, thermo-abrasion at sea level plays a central role on longer time scales and especially in initiating erosion (Kanevskiy et al., 2016). Energy-limited mass wasting appears to have been important from mid-July to late August, as a uniform or slowly decreasing activity was typical. Such mass loss driven by the energy available for ablation has previously been found to govern sub-seasonal rates of headwall retreat in Alaska (one slump) and on Banks Island, Canada (three slumps) (Lewkowicz, 1987; Barnhart, 2013). We observed steady mass wasting at the majority of slumps in the Tuktoyaktuk coastlands (C1), and also at coastal slump cliffs on the Bykovsky Peninsula. A strong link between sub-seasonal ablation rates on coastal yedoma cliffs and the energy balance has previously been observed on Muostakh Island (15 km south of Bykovsky Peninsula) (Günther et al., 2015). Our observations of rapid ablation corroborate the notion that when ice-rich permafrost is exposed above ice-poorer units, its loss is the dominant



factor in coastal terrain modification (Günther et al., 2013). By contrast, thermo-abrasion at sea level plays a central role on longer time scales and especially in initiating erosion (Kanevskiy et al., 2016).

However, deviations from energy-limited mass wasting were found also in the second half of summer, as increased activity corresponded to rainfall events for certain slumps. We believe the precipitation-linked volume losses of slumps in cluster C2
(Tuktoyaktuk coastlands) not to be a measurement artefact, as the magnitudes were large compared to the biases that changing soil moisture conditions are expected to induce (cf. Sec. S1.2). Whether precipitation really was the driver is difficult to say because of the short time series, the large study area and the lack of agreement among the available precipitation records. A precipitation-induced speed-up is certainly plausible as strong rainfall has been linked with the initiation and acceleration of thaw slump activity on a range of time scales (Burn and Friele, 1989; Balser et al., 2014; Kokelj et al., 2015). On the 11-day
time scale studied here, it could effect increased volume losses by flowing water delivering extraneous energy to the headwall (Barnhart, 2013; McRoberts and Morgenstern, 1974), or by water removing insulating debris accumulation from the headwall and evacuating accumulated sediments from the foot of the headwall and the scar zone (Burn and Friele, 1989; Barnhart, 2013; Kokelj et al., 2015). However, partitioning these effects is difficult as the sensor resolution did not allow us to distinguish headwall retreat from the evacuation of material from the foot of the headwall. A link between headwall mass wasting and
scar-zone processes is suggested by the commonly observed height changes in the scar zone. However, the temporal patterns in the scar zones were unlikely mono-causal due to their heterogeneity (e.g. Fig. S13). Certain slump floors showed increased accumulation (positive height changes) during the periods of increased headwall volume losses while others did not, indicating a non-uniform balance between the increased sediment supply and the increased capacity for removal following intense rainfall.

The late speed-up characteristic for cluster C3 is also at odds with energy-limited volume losses. Instead, it may point to
an increased sensitivity to warming as the warm season becomes longer. The mechanisms are not clear, especially considering that this phenomenon has not been commonly observed in previous studies. Distinct processes could give rise to this behaviour. It may be related to an insulating cover that persists for an uncharacteristically long time compared to the majority of thaw slumps, but rapid rates of volume losses exceeding $10 \, \mathrm{cm \, d^{-1}}$ point against the widespread applicability of this explanation. Alternatively, the acceleration could also be due to an internal instability. For instance, ice-poor parts of slump headwalls can
fail upon reaching a sufficient thaw depth or when undercut by ablating material underneath (McRoberts and Morgenstern, 1974; Wobus et al., 2011), although this also occurs earlier in the thaw period, potentially accounting for some of the earlier peaks (e.g. Fig. S11). A mechanical instability seems particularly plausible for ice-poor bluffs that do not ablate and for which a late speed-up was commonly observed (Fig. 4). However, the slightly larger peak wind speeds in late August (can, 2017) may also contribute by increasing wave-induced erosion. Irrespective of the origin, the observations highlight the need
for detailed observations and modelling efforts to better characterize the vulnerability of permafrost to warmer, longer and stormier summers.





# 6 Conclusions

This study analysed sub-seasonal dynamics of rapid permafrost thaw in two ice-rich study sites during summer 2015. Our objectives were to map thermokarst activity by observing elevation changes using single-pass interferometry and to analyse the observed sub-seasonal dynamics with respect to their spatial variability and potential drivers of permafrost degradation. Our guiding hypothesis was that mass wasting was limited by the energy required to melt the ground ice on sub-seasonal time scales, so that the 11-day mass losses should track the available energy. Our major findings and conclusions are as follows.

– The synoptic TanDEM-X single-pass observations revealed spatial variability in rates and in sub-seasonal dynamics of elevation changes which would be difficult to capture with in-situ measurements alone. The observed spatial variability was only poorly explained by macroscale characteristics such as aspect angle, which may indicate the importance of local influences such as the sub-surface ice content. In addition, observational limitations also contribute; these are induced by the small magnitude of the elevation changes, which is commonly comparable to the instrument precision, by observational biases and by the limited resolution.

– During the early thaw period in June, thaw slumps in the Tuktoyaktuk coastlands were less active than the available energy would suggest, indicating the widespread presence of an insulating veneer of debris or snow on the headwalls. In addition, a considerable amount of the ground heat flux may have to warm up the ground to the melting point before ablation can proceed more freely later in the summer.

– Later in the summer, these slumps exhibited divergent but relatively distinct patterns of volume changes. Many showed approximately uniform or slowly decreasing rates, as would be expected based on the available energy, as did the coastal thaw slump cliffs on Bykovsky, Russia. Other slumps in the Tuktoyaktuk coastlands showed pronounced and synchronous peaks, which for one type were possibly associated with strong precipitation events, coupled to removal, accumulation and transport of sediment in the scar zone. For another type, the peak occurred at the end of the thaw season. In summary, thaw slump mass wasting was not consistently limited by the available energy on approximately weekly time scales.

– The observed spatial and temporal heterogeneity of thaw slump mass wasting should be considered when predicting thermokarst rates across spatial and temporal scales. Furthermore, it has unknown consequences for the fate of the mobilized carbon, nutrients and sediments. The associated differences in exposure, lateral transport and re-burial of the thawed material deserve further attention in observational and modelling studies.

*Acknowledgements.* The authors thank Annett Bartsch, Birgit Heim and Anne Morgenstern for constructive discussions. Support by the Helmholtz Association (HA310 Remote Sensing and Earth System Dynamics) and by the European Space Agency (SP-InSARAP) is gratefully acknowledged. The TanDEM-X data were provided by DLR through proposal XTI_GEOL6759.



*Data availability.* The thaw slump inventories and the elevation change estimates have been made available https://doi.org/10.1594/PANGAEA.877506. Intermediate products described in the manuscript are available from the authors upon request. Information on how to obtain the TanDEM-X CoSSc data can be found at https://tandemx-science.dlr.de/.

*Competing interests.* The authors declare no conflict of interest.



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
