# Peer review of "Sub-seasonal thaw slump mass wasting is not consistently energy limited at the landscape scale"

_The Cryosphere, 2017_

## Referee Comment (RC1) · Anonymous Referee #1 · 12 Sep 2017

This study uses single-pass interferometry (an idea similar to DEM differencing, but in terms of interferometric phases) from bistatic TanDEM-X data to measure elevation changes at thaw slumps. Based on their results that show temporal evolutions in the summer of 2015 over two large areas in the Arctic (Tuktoyaktuk and Lena Delta), the authors quantitatively pointed out that the surface subsidence over headwalls didn't always track the changes in the input thermal energy, a conclusion that is stated in the title. This is an innovative and interesting work that I deem suitable for publishing in TC. However, I still have some comments for the authors to consider.

1. The authors pushed to the limits of the TanDEM-X data for these two landscape-scale studies on individual thaw slumps. Overall, I agree with the authors' strategies and conclusions. The authors may consider adding a summary of the following limits from the data in the discussion session.

The first limit is the spatial resolution: the multi-looked dh images have postings of 12 m, corresponding to an area limited of 1 ha, yet only 14% of the slumps are larger than 1 ha (page 8, line 28). Moreover, the analysis or interpretation is not based on individual resolution cells, but on spatially aggregated ones to active parts within each slump. I agree with this spatial-averaging approach. But I suppose this further reduces the number of slumps that can be investigated, simply because the active parts of the chosen slumps must contain several 12-m pixels. Generally speaking, would the overall results and conclusions about "not t consistently energy limited" be biased towards large slumps? Is it possible that small slumps are more likely to be energy-limited?

The second limit is the uncertainties and biases. The authors have carried out a detailed analysis on this in section 3.1.2. Spatial aggregation also helps.

The third is the limited temporal sampling. The authors produced 4 to 6 data points of temporal elevation changes for their sub-seasonal studies. This is probably the best data one can use for regional-scale mapping, thanks to the 11-day repeat of the TanDEM-X data. But there is a mismatch between the relatively poorly-sampled elevation changes and the daily meteorological changes (e.g., Fig 3). The cluster analysis indeed helps to boost the confidence level, reduce the contamination of local anomalies in individual time series, and reveal the three overall temporal patterns (Fig. 4).
2. The conclusion on "the widespread presence of an insulating veneer of debris or snow on the headwalls" is largely speculative. I understand the authors' logic and agree that this is a possible reason for the inactive phase in June. But I wish the authors can provide more direct evidence for this assertion.

3. The authors may include a clear, representative slump to illustrate the temporal changes as revealed by the TanDEM-X data, like what is presented in the supplementary materials but for a typical case. I believe this could help the readers better understand the strengths and limitations of the data as well as the key results (time series and elevation loss rate map).

4. Can the authors present the same sets of results for their two study areas? Namely, maps of slump activities and dynamics (like Fig 2a,b) for Lena River Delta, and map of elevation loss rates (like Fig 6b) for Tuktoyakuk?

5. It's not clear to me how "Time-average elevation loss rates were computed by stacking time series of individual $\Delta h$ measurements" (section 3.1.3, page 8). My understanding is that the elevation loss rates are estimated at the middle of subsequent acquisitions (Fig 3). At first, I had to guess that the authors used several pairs that have the same mid epoch but different spans to average (like two pairs day 1-day 14 and day 12-day 34 have the same mid epoch on day 23). But this would produce sparser sampling that what is shown in Figure 3. Then I found this description in S.1.1 "The stacked elevation loss rate rs was computed from the time series of estimates assuming a constant rate." Please clarify this stacking method. What is always helpful is to provide a list of images used and interferometric pairs generated and their spatial baselines (in supplementary materials).

Lastly. I felt that I reviewed two super-long papers: the main manuscript focuses on the key ideas and results centered around thaw slump geomorphology and dynamics as well as their meteorological drivers; and the supplementary material that describes the technical and mathematical details related to the measurements of elevation changes from single-pass interferometry, and clustering analysis (as well as numerous other detailed results). I understand why the authors opted this way of dividing the dense contents into two documents, esp. for TC readers. But I had to constantly go back and forth between these two documents, which greatly disrupted my reading. Practically, I found it is very difficult to include my review comments on the supplementary materials as there are not enough space and no line numbers to refer to. I provide a few comments below. I can give a more detailed review of the supplementary materials in the next round, provided that they are friendlier to reviewers.

Page S1 "The stacked elevation loss rate rs was computed from the time series of estimates ..." See my 5 comment above.
Page S2 "such a positive correlation was indeed observed (Fig. S3)". But I have to say that the positive correlation looks weak to me.
Page S7, Table S1: the units for along-track baseline should be m. And what is 'effective' baseline?

Figures related:
Figure 1e: Lack of vertical scale as the reference in the photo. Readers can have a guess from the caption though.

Figure 4, first row plots of normalized rates: clarify how the normalization is done. Why the maxima of normalized rates are not 1?

Figure 4, TDD plot: would the averaged TDD within each period be more consistent with the averaged elevation loss rates than the maximum TDD? I don't expect this would change the increasing trend in TDD though.

Figure 5a: why the elevation changes rates and their uncertainties are positively correlated?

Figure 6c. Naturally, readers expect to see close-up images for all the four boxes in 6b.

Minor comments:
Page 1, line 7: during summer *of 2015*
Page 1, line 14: the slump area *and headwall height*
Page 1, line 20: One of the motivations of this study is to advance geomorphic modeling/prediction of thermokarst. But the two papers cited (Lewkowicz, 1987; Günther et al., 2015) are both observational work, not modeling work. Please provide more situation references.
Page 2, line 5: add a comma before "which we here .."
Page 3, line 15: 'rate-limiting' is used interchangeably as 'energy-limiting'. To avoid possible confusion, change it to 'energy-limiting' or 'energy-limited'.
Page 3, line 16: the temporal signature *of volume loss*
Page 3, line 23: extra heat is used to heat and thaw the cold active layer as well.
Page 4, line 15: Shuttle Radar Topography Mission (first letters are capitalized)
Page 5, line 23: replace 'tundra lakes' with thaw lakes or thermokarst lakes
Page 6, line 29: products (plural form)
Page 6, line 30: what is the source of the input DEM? Also page 8, line 32: provide more information about the pre-disturbance DEM.
Page 7, line 12: explain what is isotropic seasonal subsidence and why it is expected to be similar at the spatial scales of your interest.

Page 8, line 3: how small? Can add the estimated magnitude as presented in the supplementary materials.

Page 9, line 3: 'earlier generation' implies multiple cycles, and more implicitly that the life cycle is about 10 years by comparing images from 2004 and 2016 . Somewhere earlier, best in the introduction, this can be mentioned.

Page 11, line 13: Fig. S8*c*, to be more specifically.

Page 11, line 29: "two peaks in mid-July and mid-August. But Fig 3d shows at least four peaks during this period. Please clarify what are the two peaks.

Page 15, line 12: "suggests a strong influence of downstream sediment dynamics.." please elaborate more on this.

Page 16, Figure 5b shows that elevation loss rates are correlated with the relief. Any comments?

Page 18, delete sentences starting from line 32 to the end of the paragraph. Same sentences appear earlier (starting from line 23). Copy/paste mistake.

Page 19, line 10: change 'effect increased' to 'increase'

---

## Referee Comment (RC2) · Anonymous Referee #2 · 30 Nov 2017

The paper " Sub-seasonal thaw slump mass wasting is not consistently energy limited at the landscape scale" is using repeated single-pass InSAR data from the TanDEM-X mission to assess the analyze the sub-seasonal thaw slump activity in two two ice-rich study sites during summer 2015. The analyzed data indicates that mass wasting in the assessed areas is not always energy limited at the landscape scale. The level of detail to which the data sets are analyzed (both scientifically and technically) is impressive. The results achieved are manifold and highly valuable for this field of research. Overall, this is an impressive paper that definitely warrants publication in this journal. That being said, I have the following comments/concerns and suggestions whose consideration might further improve the value of this paper:

Main (general) comments: 1. The paper is well written and both the description of applied methods as well as the discussions of achieved results are clear. However, the split of the material into "main paper" and "supplemental information" is not always appropriate and hinders the reading and comprehension of the material. While I understand the motivation between splitting the material into a more scientific discussion and a more technical analysis, some of the figures that are currently in the supplemental content might be better placed into the main paper to improve clarity. For instance, Figure S.19 provides a much better view of the associations between headwall elevation loss rates and slump characteristics than Figure 5b. Both figures should be grouped together and discussed together. Other figures that I would prefer in the main paper are S8, S10, and S18.

2. A major component that is currently missing in the paper is a discussion of the appropriateness of the used remote sensing data for the research at hand. I would contest that the characteristics of currently available remote sensing data such as TerraSAR-X significantly limit the information that can be extracted about thaw slump dynamics. From my point of view, the following limitations exist: 2.1 Temporal sampling: The sampling rate of 11 days seems borderline sparse given the high temporal dynamics of confounding processes such as precipitation and radiation inputs. Despite significant day-to-day variably, very little change remains when these variables are averaged over the 11-day period, making an assessment of associations difficult. 2.2 Spatial sampling: As acknowledged in various places in the paper, the 12m resolution of the InSAR-derived DEM data does not allow for a direct comparison between model outputs and surface lowering as sub-pixel variations give rise to an unknown and spatially varying scaling factor. Higher resolution would significantly improve the reliability of the remote sensing data as well as the conclusions that can be drawn based on these data. 2.3 Accuracy of surface lowering measurements: While the achieved measurement accuracy (60cm) is impressive, it is still a limiting factor especially for an analysis of processes in the scar zone, where height change rates are at the noise level.

[Figure]

It would be great to see an additional sub-section in Section 5 "Discussion" that is dedicated solely to the appropriateness of the used remote sensing resources and to suggestions for future sensors that could provide more insight into this field of research.

3. I was a bit confused by the use of the stacked elevation loss rate data (r_s) in the paper. While it is technically clear how r_s is calculated, it is not disclosed how many multi-temporal samples were used to calculate r_s. Furthermore, is it not entirely clear for which individual analyses r_s was actually employed. From my reading, I found that r_s finds very limited application in the paper and was used only once to analyze the spatial variability of the volume losses in the second half of summer. Instead, I am assuming that the elevation loss values in Figures 3 and 4 were not temporally averaged, even though this is not clearly stated in the paper. I would appreciate a clearer statement about the use of the parameter r_s in this paper.

Minor (specific) comments: 1. Page 4, line 16: Please add the following reference to the sentence ending in "in volcanology and glaciology": Kubanek J., Westerhaus, M., & Heck B. (2017). TanDEM-X time series analysis reveals lava flow volume and effusion rates of the 2012–2013 Tolbachik, Kamchatka fissure eruption. Journal of Geophysical Research: Solid Earth, 122, 7754–7774. https://doi.org/10.1002/2017JB014309

2. Page 18: Repeated identical statements seem to appear (compare lines 20 – 27 and lines 29 – line 2 on page 19). Please fix.
* * *

---

## Author Comment (AC1) · 27 Dec 2017

This study uses single-pass interferometry (an idea similar to DEM differencing, but in terms of interferometric phases) from bistatic TanDEM-X data to measure elevation changes at thaw slumps. Based on their results that show temporal evolutions in the summer of 2015 over two large areas in the Arctic (Tuktoyaktuk and Lena Delta), the authors quantitatively pointed out that the surface subsidence over headwalls didn't always track the changes in the input thermal energy, a conclusion that is stated in the title. This is an innovative and interesting work that I deem suitable for publishing in TC. However, I still have some comments for the authors to consider.

We thank the reviewer for the detailed and constructive comments, which we address point by point below. We hope that our modifications will make the manuscript clearer and more complete.

1. The authors pushed to the limits of the TanDEM-X data for these two landscape scale studies on individual thaw slumps. Overall, I agree with the authors' strategies and conclusions. The authors may consider adding a summary of the following limits from the data in the discussion session. The first limit is the spatial resolution: the multi-looked dh images have postings of 12 m, corresponding to an area limited of 1 ha, yet only 14% of the slumps are larger than 1 ha (page 8, line 28). Moreover, the analysis or interpretation is not based on individual resolution cells, but on spatially aggregated ones to active parts within each slump. I agree with this spatial-averaging approach. But I suppose this further reduces the number of slumps that can be investigated, simply because the active parts of the chosen slumps must contain several 12-m pixels. Generally speaking, would the overall results and conclusions about "not t consistently energy limited" be biased towards large slumps? Is it possible that small slumps are more likely to be energy-limited? The second limit is the uncertainties and biases. The authors have carried out a detailed analysis on this in section 3.1.2. Spatial aggregation also helps. The third is the limited temporal sampling. The authors produced 4 to 6 data points of temporal elevation changes for their sub-seasonal studies. This is probably the best data one can use for regional-scale mapping, thanks to the 11-day repeat of the TanDEM-X data. But there is a mismatch between the relatively poorly-sampled elevation changes and the daily meteorological changes (e.g., Fig 3). The cluster analysis indeed helps to boost the confidence level, reduce the contamination of local anomalies in individual time series, and reveal the three overall temporal patterns (Fig. 4). C2

We now try to provide a better account of these issues. The potential sampling bias is a good point. We now mention it explicitly in a new subsection in the discussion, along with a brief summary of the limitations of single-pass interferometry. In particular, we acknowledge the limitations imposed by the height precision (e.g. for tracking slump floor dynamics), by the spatial resolution (e.g. for distinguishing headwall retreat from processes on the slump floor), and by the temporal sampling (mentioned in the discussion of rainfall-related processes). We also tried to highlight these issues more prominently throughout the introduction ('the comparatively large measurement noise'), the methods ('As these uncertainties are comparable to the signal magnitude, a detailed uncertainty analysis is required.') and the conclusion (first bullet point).

We cannot quite follow the details of the argument about the slumps that are smaller than 1 ha. Even for smaller slumps (around Tuktoyaktuk, a typical smaller slump is around 0.3 ha in size), the resolution is

adequate for resolving those slumps: one pixel is about 100 m^2 (see Sec. 3.1.1.), a 0.3 ha slump is 3000 m^2, corresponding to 30 pixels. Clearly, a better resolution would be preferable.

2. The conclusion on "the widespread presence of an insulating veneer of debris or snow on the headwalls" is largely speculative. I understand the authors' logic and agree that this is a possible reason for the inactive phase in June. But I wish the authors can provide more direct evidence for this assertion.

We agree that our interpretation remains speculative because our data cannot separate the two processes that are most likely implicated in the observed subdued activity. These are i) a debris/snow cover and ii) heat flow into the cold slump material, and as i) reduces the energy input into the cold slump, the two are not independent. To cleanly quantify the joint role of the two processes, in-situ observations such as temperature profiles would likely be required. Unfortunately, we could not collect such data in 2015. We plan to instrument several slumps in the Tuktoyaktuk coastlands in 2018 to answer this question.

Despite the uncertainties, it is important to point out that the prevalence of an early-season veneer has been reported regularly by fieldworkers. We cite two relevant papers (Lacelle et al. 2015, Lewkowicz 1987).

To better highlight these limitations, we have extended the discussions by explicitly acknowledging the impossibility of attribution). We also draw attention to it in the conclusions.

3. The authors may include a clear, representative slump to illustrate the temporal changes as revealed by the TanDEM-X data, like what is presented in the supplementary materials but for a typical case. I believe this could help the readers better understand the strengths and limitations of the data as well as the key results (time series and elevation loss rate map).

We agree that this is a helpful addition. We have now added the elevation changes at one slump to Fig. 3 (image and time series). It is representative in that its elevation losses are very small at the beginning of summer and then pick up in mid-July. We also refer to it repeatedly in the results and discussions..

4. Can the authors present the same sets of results for their two study areas? Namely, maps of slump activities and dynamics (like Fig 2a,b) for Lena River Delta, and map of elevation loss rates (like Fig 6b) for Tuktoyakuk?

We have tried to homogenize the presentation of the results, but slight discrepancies remain. The Lena River Delta (Kurungnakh) region now includes time series of meteorological forcing to make it comparable to the other study regions. The Tuktoyaktuk coastland figure (Fig. 2) now includes a map of the seasonal elevation losses r_s, albeit not the TanDEM-X derived image itself (the slumps would be much too small to see as the extent of the study region is > 100 km), but using a colour-coded point plot. In Bykovsky, by contrast, the slumps are larger and the study area much smaller, so that we can show a map of the elevation changes for the entire study area. The dynamics, which are shown in Fig. 6b) are similar for all slumps (hence no clustering and no separate plot of cluster membership; we now mention this explicitly in the text).

5. It's not clear to me how "Time-average elevation loss rates were computed by stacking time series of individual h measurements" (section 3.1.3, page 8). My understanding is that the elevation loss rates are estimated at the middle of subsequent acquisitions (Fig 3). At first, I had to guess that the authors used several pairs that have the same mid epoch but different spans to average (like two pairs day 1-day 14 and day 12-day 34 have the same mid epoch on day 23). But this would produce sparser sampling that what is shown in Figure 3. Then I found this description in S.1.1 "The stacked elevation loss rate rs was computed from the time series of estimates assuming a constant rate." Please clarify this stacking method. What is always helpful is to provide a list of images used and interferometric pairs generated and their spatial baselines (in supplementary materials).

We have tried to clean up our descriptions:

- The methods section has been rewritten: in particular, the sub-seasonal rates r estimated from successive image pairs are explicitly contrasted with the stacked seasonal rate estimates r_s

- The caption of Fig. 3 mentions explicitly that we plotted the rate halfway between the two successive image pairs (shown on the horizontal axis)

- The temporal extent is now explicitly mentioned in all captions

- We now explicitly mention the number of TanDEM-X acquisitions in the results as well as that the results belonging to Fig. 3 were computed based on r

Lastly. I felt that I reviewed two super-long papers: the main manuscript focuses on the key ideas and results centered around thaw slump geomorphology and dynamics as well as their meteorological drivers; and the supplementary material that describes the technical and mathematical details related to the measurements of elevation changes from single-pass interferometry, and clustering analysis (as well as numerous other detailed results). I understand why the authors opted this way of dividing the dense contents into two documents, esp. for TC readers. But I had to constantly go back and forth between these two documents, which greatly disrupted my reading. Practically, I found it is very difficult to include my review comments on the supplementary materials as there are not enough space and no line numbers to refer to. I provide a few comments below. I can give a more detailed review of the supplementary materials in the next round, provided that they are friendlier to reviewers.

We apologize for the omission of line numbers. We are grateful for the comments on the supplement, which we address below.

We are aware of the large amount of information contained in the submission. However, we felt the uncertainty analyses/detailed methods (such as developments of statistical tests) did not warrant a separate submission. At the same time, we feel it is important to describe certain technical aspects like the biases in detail (not least because certain analyses are, as the reviewer rightly contends, close to the technical limits imposed by the technique).

To improve the flow of the paper we have moved results-related figures from the supplement to the main body of the document (cluster statistics and membership). We hope that this facilitates reading the manuscript without recourse to the supplement.

Page S1 "The stacked elevation loss rate rs was computed from the time series of estimates ..."
See my 5 comment above.

We address the description of r_s above.

Page S2 "such a positive correlation was indeed observed (Fig. S3)". But I have to say that the positive correlation looks weak to me.

Agreed. We now qualify the strength of the correlation in the text as weak.

Page S7, Table S1: the units for along-track baseline should be m. And what is 'effective' baseline?

To clarify these issues, we have made two changes. We have renamed the quantity to along-track interferometric time lag. The advantage over the spatial baseline is that it is much easier to interpret. We have also cited a relevant paper (Suchandt and Runge; in both the text and the caption) that discusses the concept of effective time separation (or baseline) in detail: loosely speaking it is the time separation so that the along-track phase phi = 2*k*v_los*tau_eff, where v_los is the line of sight velocity of the target. It differs from the standard concept for ATI systems in which one of the antennas is purely passive.

Figures related: Figure 1e: Lack of vertical scale as the reference in the photo. Readers can have a guess from the caption though.

We have considered adding a scale to the picture itself, but we think that the caption suffices.

Figure 4, first row plots of normalized rates: clarify how the normalization is done. Why the maxima of normalized rates are not 1?

We now mention that the absolute elevation changes sum to one. This is the same normalization we use in the tests (Eq. S4).

Figure 4, TDD plot: would the averaged TDD within each period be more consistent with the averaged elevation loss rates than the maximum TDD? I don't expect this would change the increasing trend in TDD though.

The reason we show the maximum TDD is the potential link between the end-of-summer acceleration to a crossing of a thaw depth threshold, which in turn would be related to the maximum TDD. We mention the crossing of a thaw depth threshold in Sec. 5.

Figure 5a: why the elevation changes rates and their uncertainties are positively correlated?

Good point, to which we do not have a compelling answer. We believe at least two factors to be important. First, larger headwalls tend to have larger volume losses, but they also are prone to certain uncertainty increasing measurement artefacts such as radar shadow, cf. discussion of uncertainties. Second, selection biases will reinforce this effect, as slumps that are, for instance, small and stable would have a high uncertainty and a low volume loss rate, and would thus less likely show detectable activity.

Figure 6c. Naturally, readers expect to see close-up images for all the four boxes in 6b.

We realize this is not ideal. However, we tried shrinking the insets so we could include all four but found the previous version to be more effective.

Minor comments:

Page 1, line 7: during summer *of 2015*

done

Page 1, line 14: the slump area *and headwall height*

We have replaced slump area with headwall height

Page 1, line 20: One of the motivations of this study is to advance geomorphic modeling/prediction of thermokarst. But the two papers cited (Lewkowicz, 1987; Günther et al., 2015) are both observational work, not modeling work. Please provide more situation references.

We have included Westermann et al. as an example for a mechanistic (as opposed to data-driven) model

Page 2, line 5: add a comma before "which we here .."

done

Page 3, line 15: 'rate-limiting' is used interchangeably as 'energy-limiting'. To avoid possible confusion, change it to 'energy-limiting' or 'energy-limited'.

done (but slight rephrasing)

Page 3, line 16: the temporal signature *of volume loss*

done

Page 3, line 23: extra heat is used to heat and thaw the cold active layer as well.

Good point. The next sentence now reads: 'Early summer mass wasting may also be subdued because the incoming energy is used to warm the cold permafrost to the melting point before ablation can set in'

Page 4, line 15: Shuttle Radar Topography Mission (first letters are capitalized)

done

Page 5, line 23: replace 'tundra lakes' with thaw lakes or thermokarst lakes

We have replaced 'tundra lakes' with 'lakes'; the reason for not specifying the lakes further is their geological diversity (glacial processes have likely contributed to shaping especially some of the larger lakes)

Page 6, line 29: products (plural form)

Rephrased sentence

The input DEM was derived from TanDEM-X data acquired before or during the Science Phase.

Also page 8, line 32: provide more information about the pre-disturbance DEM.

We now provide details on the DEM (MVAP DEM, 2008, Northwest Territories Centre for Geomatics) in the text.

Page 7, line 12: explain what is isotropic seasonal subsidence and why it is expected to be similar at the spatial scales of your interest.

We now refer to the subsidence associated with top-down thaw as permafrost thaw subsidence and provide a citation. The attribute 'isotropic' is commonly employed to describe the spatially uniform nature of the subsidence.

Page 8, line 3: how small? Can add the estimated magnitude as presented in the supplementary materials.

2 cm vs. 30-60 cm; now both sets of numbers are included in the main document, as is a reference to the appropriate Fig. in the supplement

Page 9, line 3: 'earlier generation' implies multiple cycles, and more implicitly that the life cycle is about 10 years by comparing images from 2004 and 2016 . Somewhere earlier, best in the introduction, this can be mentioned.

We have added two sentences on polycyclic activity in the introduction

Page 11, line 13: Fig. S8*c*, to be more specifically.

Done; the figure has been moved to the main document.

Page 11, line 29: "two peaks in mid-July and mid-August. But Fig 3d shows at least four peaks during this period. Please clarify what are the two peaks.

The two peaks refer to the restricted time period shown in Fig. 4; we now reference Fig. 4 explicitly

Page 15, line 12: "suggests a strong influence of downstream sediment dynamics.." please elaborate more on this.

Done, but in the discussions. There, we discuss sediment supply driven by mass wasting at the headwall, and sediment evacuation from the scar zone. We also mention that our observations are insufficient for resolving these processes.

Page 16, Figure 5b shows that elevation loss rates are correlated with the relief. Any comments?

We mention it in the text but do not discuss it in detail. Apart from the direct increase of volume losses with headwall relief that would be expected if the planimetric backwasting was constant, there is also evidence for increased backwasting at larger slumps (Lacelle et al. 2015), albeit at longer time scales.

Page 18, delete sentences starting from line 32 to the end of the paragraph. Same sentences appear earlier (starting from line 23). Copy/paste mistake.

done

Page 19, line 10: change 'effect increased' to 'increase'

done

---

## Author Comment (AC2) · 27 Dec 2017

The paper " Sub-seasonal thaw slump mass wasting is not consistently energy limitedat the landscape scale" is using repeated single-pass InSAR data from the TanDEM-Xmission to assess the analyze the sub-seasonal thaw slump activity in two two ice-rich study sites during summer 2015. The analyzed data indicates that mass wasting in the assessed areas is not always energy limited at the landscape scale. The level of detail to which the data sets are analyzed (both scientifically and technically) is impressive.The results achieved are manifold and highly valuable for this field of research. Overall,this is an impressive paper that definitely warrants publication in this journal.

We are grateful to the referee for their helpful and constructive comments. We believe that thanks to the reviewer's input the clarity and quality of the revised manuscript have improved.

That being said, I have the following comments/concerns and suggestions whose consideration might further improve the value of this paper:

Main (general) comments:

1. The paper is well written and both the description ofapplied methods as well as the discussions of achieved results are clear. However,the split of the material into "main paper" and "supplemental information" is not always appropriate and hinders the reading and comprehension of the material. While I understand the motivation between splitting the material into a more scientific discussion and a more technical analysis, some of the figures that are currently in the supplemental content might be better placed into the main paper to improve clarity. For instance, FigureS.19 provides a much better view of the associations between headwall elevation loss rates and slump characteristics than Figure 5b. Both figures should be grouped together and discussed together. Other figures that I would prefer in the main paper are S8, S10, and S18.

We hope we have now struck a better balance between the main document and the supplement. We have moved Fig. S8 and S18 in adapted form into the main body of the manuscript. The reason for including these two figures is that they deal with the process-based results (focus of the paper), whereas Fig. S10 is more to do with technical issues. The reason for not including S19 is that we focus on the sub-seasonal mass wasting, whereas S19 deals with longer time scales. We believe Fig. 5b provides sufficient information for our purposes; in particular, the regression analysis accounts for the fact that the explanatory variables like aspect and area are correlated. In addition, we have tried to provide a better link between the main document and the supplementary material.

2. A major component that is currently missing in the paper is a discussion of the appropriatenessof the used remote sensing data for the research at hand. I would contest that the characteristics of currently available remote sensing data such as TerraSARX significantly limit the information that can be extracted about thaw slump dynamics. From my point of view, the following limitations exist:

2.1 Temporal sampling: The sampling rate of 11 days seems borderline sparse given the high temporal dynamics of confounding processes such as precipitation and radiation inputs. Despite significant day-to-day variably, very little change remains when these variables are averaged over the 11-day period, making an assessment of associations difficult.

2.2 Spatial sampling: As acknowledged in various places in the paper, the 12m resolution of the InSAR-derived DEM data does not allow for a direct comparison between model outputs and surface lowering as sub-pixel variations give rise to an unknown and spatially varying scaling factor. Higher resolution would significantly improve the reliability of the remote sensing data as well as the conclusions that can be drawn based on these data.

2.3 Accuracy of surface lowering measurements: While the achieved measurement accuracy (60cm) is impressive, it is still a limiting factor especially for an analysis of processes in the scar zone, where height change rates are at the noise level.

It would be great to see an additional sub-section in Section 5 "Discussion" that is dedicated solely to the appropriateness of the used remote sensing resources and to suggestions for future sensors that could provide more insight into this field of research.

We agree with the referee and have tried to paint a clearer picture. To this end, we have added an additional subsection in the discussion. There we briefly summarize the key limitations, while also highlighting the technique's advantages and contrasting them with complementary tools such as LiDAR. We also acknowledge the uncertainties and biases throughout the introduction (e.g. 'the comparatively large measurement noise'), the methods ('As these uncertainties are comparable to the signal magnitude, a detailed uncertainty analysis is required.') and the conclusions (first bullet point).

We also pick up on existing and future observing systems that hold promise, drawing attention e.g. to the potential of higher radar frequencies such as Ku band, as they can achieve higher spatial resolutions and accuracies.

3. I was a bit confused by the use of the stacked elevation loss rate data (r_s) in thepaper. While it is technically clear how r_s is calculated, it is not disclosed how many multi-temporal samples were used to calculate r_s. Furthermore, is it not entirely clearfor which individual analyses r_s was actually employed. From my reading, I found that r_s finds very limited application in the paper and was used only once to analyze the spatial variability of the volume losses in the second half of summer. Instead, I am assuming that the elevation loss values in Figures 3 and 4 were not temporarily averaged, even though this is not clearly stated in the paper. I would appreciate a clearer statement about the use of the parameter r_s in this paper.

We have better highlighted the temporal extent of the elevation loss rate observations throughout the manuscript. The temporal extent is now explicitly mentioned in all captions, and we have also clarified the result sections (e.g. mentioning also the number of TanDEM-X acquisitions, and highlighting that Fig 3 refers to non-averaged rates computed from successive image pairs). Finally, we have slightly rewritten the description of the computation of r_s in the methods, highlighting its purpose (visualization, spatial comparison; Fig. 5) and contrasting it with the computation of the subseasonal rates between successive image pairs, which we now refer to as r throughout the manuscript.

Minor (specific) comments:

1. Page 4, line 16: Please add the following reference to the sentence ending in "in volcanology and glaciology": Kubanek J., Westerhaus, M., & Heck B. (2017). TanDEM-X time series analysis reveals lava flow volume and effusion rates of the 2012–2013 Tolbachik, Kamchatka fissure eruption. Journal of Geophysical Research: Solid Earth, 122, 7754–7774. https://doi.org/10.1002/2017JB0143092.

done

Page 18: Repeated identical statements seem to appear (compare lines 20 – 27and lines 29 – line 2 on page 19). Please fix.

done